# POLISH: Adaptive Online Cross-Modal Hashing for Class Incremental Data

## ABSTRACT

In recent years, hashing-based online cross-modal retrieval has garnered growing attention. This trend is motivated by the fact that web data is increasingly delivered in a streaming manner as opposed to batch processing. Simultaneously, the sheer scale of web data sometimes makes it impractical to fully load for the training of hashing models. Despite the evolution of online cross-modal hashing techniques, several challenges remain: 1) Most existing methods learn hash codes by considering the relevance among newly arriving data or between new data and the existing data, often disregarding valuable global semantic information. 2) A common but limiting assumption in many methods is that the label space remains constant, implying that all class labels should be provided within the first data chunk. This assumption does not hold in real-world scenarios, and the presence of new labels in incoming data chunks can severely degrade or even break these methods.

To tackle these issues, we introduce a novel supervised online cross-modal hashing method named adaPtive Online cLass-Incremental haSHing (POLISH). Leveraging insights from language models, POLISH generates representations for new class label from multiple angles. Meanwhile, POLISH treats label embeddings, which remain unchanged once learned, as stable global information to produce high-quality hash codes. POLISH also puts forward an efficient optimization algorithm for hash code learning. Extensive experiments on two real-world benchmark datasets show the effectiveness of the proposed POLISH for class incremental data in the cross-modal hashing domain.

**ACM Reference Format:**
Anonymous Author(s). 2018. POLISH: Adaptive Online Cross-Modal Hashing for Class Incremental Data. In *Proceedings of The Web Conference (Conference acronym 'XX)*. ACM, New York, NY, USA, 10 pages. https://doi.org/XXXXXXX.XXXXXXX

## 1 INTRODUCTION

Due to the explosive growth of heterogeneous modalities web data, the task of cross-modal search in large datasets has evolved into a significant challenge. Traditional search methods are no longer the optimal choice in these scenarios due to time and storage complexities. To address this challenge, approximate nearest-neighbor (ANN) search methods, particularly those based on learning to hash [5, 23, 29–31], have gained substantial attention in recent years. By performing bitwise XOR operations in the Hamming space of hash

codes, these methods enable efficient searches within the binary encodings. Moreover, the short binary codes generated from mapping high-dimensional data still retain the underlying similarity between samples in the original space [37].

Cross-modal hashing methods have achieved significant performance [7, 24, 34], thanks to large datasets and substantial memory resources. However, this poses significant challenges for the application of intelligent agents, as we need to learn new knowledge from continuously arriving training samples under resource constraints. When new data appears, batch-based cross-modal hashing requires accumulating both new and old data to form a new database and retraining mapping functions. This approach not only incurs computational costs but also poses storage challenges as data streams. Consequently, the challenging scenario of learning from online multi-modal data streams is introduced [26, 39]. After completing one round of learning, online cross-modal hashing typically removes the data of this round from memory, making it unavailable for future access. This strategy allows the online methods to efficiently accommodate continuously incoming data chunks. However, there are still some issues to consider. 1) A notable challenge lies in effectively leveraging label information to enhance the encoding capacity of binary hash codes. Current cross-modal online methods typically update hash functions based on the relevance between newly arrived data or new data and existing data, often overlooking global information. Even the one paper [13] considers using co-occurrence correlation to capture label inter dependencies, but it fails to account for the rich semantic information inherent in labels. 2) How to enhance the adaptability of the model to accommodate incremental label space. Thus far, the majority of existing online methods implicitly assume that the label space is static, meaning all class labels should be provided in the initial data chunk. In practice, this assumption might be unrealistic. When new labels surface in newly arriving data chunks, these methods may struggle to effectively update their hash functions.

To address the aforementioned challenges, we introduce a novel supervised cross-modal online hashing method in this paper, named adaPtive Online cLass-Incremental haSHing (POLISH). Although online cross-modal retrieval has achieved satisfactory performance, it encounters difficulties in effectively handling the issue of incremental class labels that emerge with the arrival of new data. Consequently, POLISH offers an adaptive approach, capable of seamless integration with existing online cross-modal hashing methods to dynamically accommodate the expansion of label spaces. As shown in Figure 1, POLISH leverages category correlation and semantic information obtained by language models to generate representations for new class label. It incorporates hash code characteristics into label embeddings using Hadamard matrices. Simultaneously, POLISH employs label embeddings as globally invariant information to guide the generation of high-quality hash codes for new data chunks. Additionally, POLISH introduces an efficient optimization algorithm for discrete learning of hash codes and label embeddings.

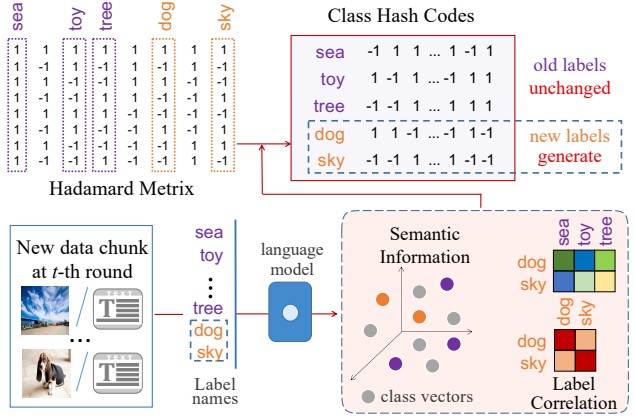

**Figure 1: The framework of the proposed POLISH. POLISH leverages category correlation and semantic information obtained by language models to generate embeddings for new class labels. Simultaneously, it incorporates hash code characteristics into label embeddings using Hadamard matrices.**

The contributions of this paper are summarized as follows:

- We introduce a novel adaptive supervised online cross-modal hashing method. By leveraging the multi-faceted information provided by language models, we learn label embeddings, which represent global information. Meanwhile, these embeddings guide the learning process, enabling POLISH to generate more accurate hash codes.
- To the best of our knowledge, this is the first explicit attempt to design a method capable of accommodating incremental label space in the domain of online cross-modal hashing.
- We propose a discrete online optimization algorithm that allows for the discrete learning of binary codes while maintaining binary constraints.
- We conduct extensive experiments on widely-used benchmark datasets, which demonstrate the superiority of the proposed POLISH for class incremental data in the online cross-modal hashing domain. Besides, We will release the code for POLISH soon and hope that it could facilitate other researchers and the community.

## 2 RELATED WORK

### 2.1 Online Cross-Modal Hashing

Existing cross-modal hashing can be broadly categorized into two main classes: unsupervised and supervised methods. In unsupervised methods, cross-modal correlations between heterogeneous data are explored within their original space, without leveraging any supervised information. Representative unsupervised methods include Collective Matrix Factorization Hashing (CMFH) [4], Composite Correlation Quantization (CCQ) [22], and others. Conversely, supervised methods incorporate semantic information to guide the learning of hash codes, thereby enhancing performance. Notable supervised methods encompass Discrete Cross-modal Hashing (DCH) [38], Label Consistent Matrix Factorization Hashing (LCMFH) [32], and Scalable Discrete Matrix Factorization Hashing (SCRATCH) [14], among others. In recent times, numerous

end-to-end deep cross-modal hashing methods have emerged, like Deep Cross-Modal Hashing (DCMH) [11] and Self-Supervised Adversarial Hashing (SSAH) [15]. These methods integrate feature extraction and hash code learning within a single framework.

While current cross-modal hashing approaches have demonstrated commendable performance, most of them are designed for batch processing, which entails learning mapping functions using fixed, non-expanding datasets. In real-world scenarios, however, multiple modal data often arrives in a streaming fashion. This can pose several challenges for computational and memory costs. Thus, the need for online cross-modal hashing is increasing [6, 9, 18]. Online cross-modal hashing methods can be classified into two main categories: unsupervised hashing and supervised hashing. Unsupervised online cross-modal methods, such as Online Collective Matrix Factorization Hashing (OCMFH) [33], learn hash codes and functions by capitalizing on inherent data properties. Supervised online cross-modal hashing takes advantage of label information to guide the learning process [16, 20, 40]. For instance, Online Latent Semantic Hashing (OLSH) [39] maps discrete labels into a continuous latent semantic concept space, using this space to steer hash-code learning. Online Cross-Modal Scene Retrieval (OCMSR) [26] constructs a semantic graph by mapping text and images to a graph and assesses the similarity between hash codes of different modalities using cosine metrics. Label Embedding Online Hashing (LEMON) [35] updates hash functions based on the correlation between newly arriving data and existing data. Discrete Online Cross-modal Hashing (DOCH) [44] is based on a discrete latent factor model that directly measures the similarity between new and old data in the Hamming space. It also introduces an efficient optimization algorithm for discrete hash code learning. Label-Semantic-Enhanced Online Hashing (LSE-OH) [13] captures the similarity between samples and the underlying dependency between labels and utilizes them to learn discrete hash codes. However, these methods all assume a fixed label space. If new labels arrive with new data, they may perform poorly or even fail to work.

### 2.2 Hashing Methods with Language Model

In recent years, there have been significant advancements in the field of vision and language models, particularly in the context of multi-modal tasks. Cross-modal hashing methods are designed to facilitate cross-modal retrieval services for data in the image and text modalities. These methods seamlessly integrate with state-of-the-art visual and semantic models, such as CLIP. Consequently, some researchers have leveraged CLIP to enhance cross-modal hashing. The specific strategies employed include: 1) Harnessing CLIP's powerful representational capabilities to generate features for both image and text data [36, 41]. 2) Utilizing CLIP's knowledge to facilitate alignment between image and text modalities [47].

In recent months, there has been a surge of interest and attention directed towards large-scale language models. These models have demonstrated exceptional performance and robust generalization capabilities. They can effectively tackle a wide range of tasks and data distributions, even those that extend beyond the specific data they were trained on. In this work, we use the prior knowledge extracted from language models to address class-incremental challenges, thereby enhancing the performance of retrieval tasks.

## 3 POLISH

In this section, we briefly introduce our method. First, we generate the representation for each class label with the help of language models and the Hadamard matrix. Then, we use learned label embedding to generate high-quality hash codes of samples. The following is a detailed description of each module.

### 3.1 Notations and Problem Description

*3.1.1 Notations.* In this paper, we use mathematical notation conventions as follows: Boldface lowercase letters, e.g., $\mathbf{a}$, represent vectors, while boldface uppercase letters, e.g., $\mathbf{A}$, represent matrices. $\mathbf{A}_i$ denotes the $i$-th column of matrix $\mathbf{A}$. The transpose of $\mathbf{A}$ is denoted as $\mathbf{A}^\top$, and its inverse is represented as $\mathbf{A}^{-1}$. The Frobenius norm of a vector or matrix is denoted as $\| \cdot \|_F$, where $\| \mathbf{A} \|_F^2 = Tr(\mathbf{A}^\top \mathbf{A})$, with $Tr(\cdot)$ signifying the trace of a square matrix. We also use $\mathbf{I}$ for the identity matrix, $\mathbf{1}$ for an all-one matrix, and $\mathbf{0}$ for an all-zero matrix.

Now, considering the scenario of streaming training data composed of $M$ modalities, we introduce the notations. At the $t$-th round, a new data chunk $\vec{\mathbf{X}}^{(t)}m \in \mathbb{R}^{d_m \times n_t}$ is added to the training set, along with class labels $\vec{\mathbf{L}}^{(t)} \in \{0,1\}^{(C_{t-1}+c_t) \times n_t}$, where $m = 1, ..., M$ indicates the modality index, $n_t$ represents the size of the new data chunk, $d_m$ is the dimensionality of features in the $m$-th modality, and $c_t$ is the number of newly introduced labels at the $t$-th round. $C_{t-1} = \sum_{i=1}^{t-1} c_i$ is the count of existing classes up to round $t-1$.

It is essential to highlight that we explicitly define $c_t$ and $C_{t-1}$ to illustrate that our proposed method does not make an assumption that the label space is static, meaning not all class labels need to be provided in the initial data chunk. In Section 3.2, we will discuss how to manage situations where new labels appear in subsequent data chunks. Correspondingly, the existing data accumulated prior to round $t$ is denoted as $\tilde{\mathbf{X}}_m^{(t)} \in \mathbb{R}^{d_m \times N_{t-1}}$, with $N_{t-1} = \sum_{i=1}^{t-1} n_i$ representing the size of this existing dataset. The corresponding label matrix for this existing data is denoted as $\tilde{\mathbf{L}}^{(t)} \in \{0,1\}^{C_{t-1} \times N_{t-1}}$.

*3.1.2 Problem Description.* When a new data chunk arrives in round $t$, the hash code representing the entire data chunk is denoted as $[\tilde{\mathbf{B}}^{(t)}, \vec{\mathbf{B}}^{(t)}] \in \{-1,1\}^{r \times N_t}$, where $r$ signifies the length of the hash code, $\tilde{\mathbf{B}}^{(t)} \in \{-1,1\}^{r \times N_{t-1}}$ represents the hash code for the previously existing data. $\vec{\mathbf{B}}^{(t)} \in \{-1,1\}^{r \times n_t}$ is the hash code for the newly introduced data.

The primary objectives of our method are as follows: 1) Training the model effectively in a scenario where new classes may continuously emerge alongside new data chunks. 2) Generating $r$-bit binary codes $\vec{\mathbf{B}}^{(t)}$ to represent the newly arriving data while keeping the hash codes of the existing data, denoted as $\tilde{\mathbf{B}}^{(t)}$, unchanged.

### 3.2 Label Embedding with Language Model

Language models are normally pretrained through self-supervised learning on large-scale text corpora. During this process, the model is tasked with predicting the next word or a segment of text given a context. This forces the language model to learn the semantics and contextual information of the text. Thanks to its robust generalization and contextual comprehension abilities, it has demonstrated impressive performance in various applications such as text classification, text generation, cross-modal tasks, and more. In this paper, we utilize it to assist us in generating learnable embeddings for labels, serving as global guidance for the model. Specifically, we utilize widely-used language models, e.g., Word2Vec[25], BERT[3], CLIP[27], and BLOOM [28], to transform label words into embeddings, referred to as $\mathbf{E} \in \mathbb{R}^{d_e \times c}$. Nevertheless, these embeddings cannot be employed directly for hash code learning for two primary reasons. 1) The dimensionality of these embeddings does not align with that of hash codes. 2) Embeddings lack learnability and do not meet the characteristics of hash codes, including bit balance and maximal information entropy. Consequently, we utilize $\mathbf{E}$ to generate fresh embeddings for label words, focusing on two aspects: preserving category correlation relationships and upholding category semantic information.

*3.2.1 Preserving Label Correlation Relationships.* Maintaining correlation relationships among categories is crucial for retrieval tasks. This practice helps cluster similar multi-modal data points together, ultimately improving retrieval efficiency. Furthermore, the preservation of correlation relationships among categories enhances the model's ability to generalize to streaming data effectively. When multi-modal samples of the same category maintain similarities in feature space, the method can more easily generalize to newly incoming multi-modal samples. Inspired by the most commonly used terms in the hashing domain for embedding pairwise similarity of data, i.e., $\|r\mathbf{S} - \mathbf{B}^T\mathbf{B}\|_F^2$, we design the following loss to learn the embedding of class labels $\mathbf{R}$. More specifically, at the first round, $\mathbf{R}^{(1)}$ is embedded in the following loss,

$$\min_{\mathbf{H}^{(1)}} \| r\mathbf{A}^{(1)} - \mathbf{R}^{(1)\top}\mathbf{R}^{(1)} \|_F^2, \tag{1}$$

where $\mathbf{R}^{(1)} \in \mathbb{R}^{r \times C_1}$ is the real-valued embedding of class labels at the first round. $\mathbf{A}^{(1)} \in \mathbb{R}^{C_1 \times C_1}$ is the pairwise correlation among $C_1$ labels at the first round. It is computed by $\mathbf{A}^{(1)} = cos(\mathbf{E}^{(1)}, \mathbf{E}^{(1)})$, where $cos()$ is the Cosine Similarity.

We further emphasize that our method is explicitly designed to tackle the challenging scenario of learning from online data streams, even when never-seen-before classes are encountered. Thus, when the $t$-th new data chunk arrives, two possible situations may arise. In the first situation, there are no new classes introduced, indicated by $c_t = 0$ and $C_{t+1} = C_t$. In this case, all embeddings of existing labels already have been learned, and we set $\mathbf{R}^{(t-1)} = \mathbf{R}^{(t)}$. The second situation occurs when $c_t$ new classes are introduced. In such a situation, we proceed to learn label embeddings for these new classes, which can be formulated as $\vec{\mathbf{R}}^{(t)} \in \mathbb{R}^{r \times c_t}$. It is worth noting that the label embeddings learned in previous rounds remain unchanged, that is, $\tilde{\mathbf{R}}^{(t)} = [\tilde{\mathbf{R}}^{(t-1)}, \vec{\mathbf{R}}^{(t-1)}]$, where $\tilde{\mathbf{R}}^{(t)} \in \mathbb{R}^{r \times C_{t-1}}$ denotes label embeddings of the existing labels in the $t$-th round. To generate label embeddings for new classes, we first extend the formulation of $\mathbf{A}^{(t)}$ in round t,

$$\mathbf{A}^{(t)} = \begin{bmatrix} \mathbf{A}_{oo}^{(t)} & \mathbf{A}_{on}^{(t)} \\ \mathbf{A}_{no}^{(t)} & \mathbf{A}_{nn}^{(t)} \end{bmatrix}, \tag{2}$$

where,

$$\mathbf{A}_{oo}^{(t)} = cos(\mathbf{E}^{\tilde{(t)}}, \mathbf{E}^{\tilde{(t)}}) \quad \mathbf{A}_{on}^{(t)} = cos(\mathbf{E}^{\tilde{(t)}}, \mathbf{E}^{\vec{(t)}})$$
$$\mathbf{A}_{no}^{(t)} = cos(\mathbf{E}^{\vec{(t)}}, \mathbf{E}^{\tilde{(t)}}) \quad \mathbf{A}_{nn}^{(t)} = cos(\mathbf{E}^{\vec{(t)}}, \mathbf{E}^{\vec{(t)}}) \tag{3}$$

$\mathbf{E}^{\tilde{(t)}} \in \mathbb{R}^{d_e \times C_{t-1}} = \mathbf{E}^{(t-1)}$ is the embeddings of old classes generated by language models, $\mathbf{E}^{\vec{(t)}} \in \mathbb{R}^{d_e \times c_t}$ is the embeddings of new classes (classes which first appear at the $t$-th round) generated by language models, and $cos()$ is the Cosine Similarity. Then, by substituting $\mathbf{A}^{(t)}$ and $[\tilde{\mathbf{R}}^{(t)}, \vec{\mathbf{R}}^{(t)}]$ into Eq.(1), the following loss function is given,

$$\min_{\vec{\mathbf{R}}^{(t)}} \| r\mathbf{A}_{on}^{(t)} - \tilde{\mathbf{R}}^{(t)\top}\vec{\mathbf{R}}^{(t)} \|_F^2 + \| r\mathbf{A}_{nn}^{(t)} - \vec{\mathbf{R}}^{(t)\top}\vec{\mathbf{R}}^{(t)} \|_F^2, \tag{4}$$

where constant terms are already omitted.

By Eq.(4), the proposed method is designed to be adaptable to class-incremental scenarios, where learning the label embeddings for new classes $\vec{\mathbf{R}}^{(t)}$, while the embeddings for old existing classes $\tilde{\mathbf{R}}^{(t)}$ remain unchanged. However, one issue that needs to be further considered is how to make the learned category representations better align with the characteristics of hash codes. To tackle this, we turn to the Hadamard matrix for assistance.

In general, the Hadamard matrix is a binary square matrix of order $2^k$, where its entries are either $-1$ or $+1$. Additionally, its row vectors and column vectors are pairwise orthogonal. Following the approach in [19], we denote the size of the Hadamard matrix as $g$ and set $g$ as follows,

$$g = \min\{l | l = 2^k, l \geq r, l \geq C_t, k = 1, 2, 3, ...\}, \tag{5}$$

where $r$ represents the length of hash codes, and $C_t$ is the number of old class labels at the $t$-th round. We proceed to construct the Hadamard matrix by defining the entry in the $i$-th row and the $j$-th column as $(-1)^{(i-1)\times(j-1)}$. In cases where the length of hash codes does not match the Hadamard matrix, meaning $r \neq g$, we apply a strategy similar to that in [19], utilizing a random Gaussian matrix to address this disparity. Consequently, we obtain Hadamard representation for all class labels, denoted as $\mathbf{H}^{(t)}$.

Then, we replace one $\mathbf{R}^{(t)}$ with Hadamard representation $\mathbf{H}^{(t)}$ in Eq.(1) and Eq.(4), which is effective to let embedding of class labels meet the characteristics of hash codes. Therefore, we have the final objective function as follows,

$$\min_{\mathbf{R}^{(1)}} \| r\mathbf{A}^{(1)} - \mathbf{H}^{(1)\top}\mathbf{R}^{(1)} \|_F^2 + \| \mathbf{H}^{(1)} - \mathbf{R}^{(1)} \|_F^2, \tag{6}$$

$$\min_{\vec{\mathbf{R}}^{(t)}} \| r\mathbf{A}_{on}^{(t)} - \tilde{\mathbf{H}}^{(t)\top}\vec{\mathbf{R}}^{(t)} \|_F^2 + \| r\mathbf{A}_{nn}^{(t)} - \vec{\mathbf{H}}^{(t)\top}\vec{\mathbf{R}}^{(t)} \|_F^2$$
$$+ \| \vec{\mathbf{H}}^{(t)} - \vec{\mathbf{R}}^{(t)} \|_F^2, \tag{7}$$

In these two formulas, the reasons for replacing $\mathbf{R}^{(t)}$ with $\mathbf{H}^{(t)}$ are: 1) Transforming the properties of the Hadamard matrix to $\mathbf{R}^{(t)}$. The Hadamard matrix, except for the first row and first column, consists of elements that are half +1 and half $-1$. This imparts the property of bit balance to $\mathbf{R}^{(t)}$. Additionally, the rows and columns of the Hadamard matrix are mutually orthogonal, endowing $\mathbf{R}^{(t)}$ with maximal information entropy. 2) Avoiding the complex optimization of $\mathbf{R}^{(t)}$ in Eq.(6) and Eq.(7).

In the online hashing domain, several Hadamard matrix-based methods have been introduced, including HCOH [19], HMOH [17], and Adaptive Online Multi-modal Hashing via Hadamard Matrix [42]. However, as far as our knowledge extends, none of these approaches fall within the scope of online cross-modal hashing. In this paper, while we drew some inspiration from Hadamard matrix-based hashing techniques and integrated the Hadamard matrix into our learning process, our proposed method is fundamentally distinct from all existing Hadamard matrix-based hashing methods.

*3.2.2 Preserving Label Semantic Information.* When learning label representations, it is crucial to consider not just the correlation relationships between categories but also their semantic information. Category semantics encompass additional contextual and meaningful details pertaining to the categories, encompassing features, attributes, and more. The inclusion of semantic information assists the model in accurately defining boundaries between categories and improve the retrieval performance. For example, considering the semantic aspect of an aircraft, like "flying" can facilitate its differentiation from other modes of transportation. Thanks to the capabilities of robust language models, semantic information can be seamlessly incorporated into embeddings. Consequently, we introduce embeddings acquired through language models into our learning process. The loss is as follows,

$$\min_{\vec{\mathbf{R}}^{(t)}} \| \tilde{\mathbf{E}}^{(t)} - \tilde{\mathbf{R}}^{(t)\top}\mathbf{W}^{(t)} \|_F^2 + \| \vec{\mathbf{E}}^{(t)} - \vec{\mathbf{R}}^{(t)\top}\mathbf{W}^{(t)} \|_F^2, \tag{8}$$

where $\mathbf{E}^{\tilde{(t)}}$ is the embeddings of old classes generated by language models, $\mathbf{E}^{\vec{(t)}}$ is the embeddings of new classes generated by language models, and $\mathbf{W}^{(t)}$ is a mapping matrix. Based on the online hashing settings, the newly arrived data and the old accumulated data should be both considered to generate $\vec{\mathbf{R}}^{(t)}$.

Jointly considering Eq.(8) and Eq.(7), we have the following objective function to learn the embedding of labels,

$$\min_{\vec{\mathbf{R}}^{(t)}} \| r\mathbf{A}_{on}^{(t)} - \tilde{\mathbf{H}}^{(t)\top}\vec{\mathbf{R}}^{(t)} \|_F^2 + \| r\mathbf{A}_{nn}^{(t)} - \vec{\mathbf{H}}^{(t)\top}\vec{\mathbf{R}}^{(t)} \|_F^2$$
$$+ \| \vec{\mathbf{H}}^{(t)} - \vec{\mathbf{R}}^{(t)} \|_F^2 + \alpha \| \tilde{\mathbf{E}}^{(t)} - \tilde{\mathbf{R}}^{(t)\top}\mathbf{W}^{(t)} \|_F^2 \tag{9}$$
$$+ \alpha \| \vec{\mathbf{E}}^{(t)} - \vec{\mathbf{R}}^{(t)\top}\mathbf{W}^{(t)} \|_F^2,$$

where $\alpha$ is the trade-off parameter. Apparently, the above loss integrates the two key components, i.e., Label Correlation Relationships and Label Semantic Information. The strategy of learning class representations $\vec{\mathbf{R}}^{(t)}$ and using them as globally invariant information to guide hash code learning has been proven effective [17, 19, 43, 46]. Since the representations based on Hadamard $\vec{\mathbf{H}}^{(t)}$ and embeddings generated by language models $\mathbf{E}^{(t)}$ are non-learnable, we prefer to utilize them as auxiliary tools. Unlike $\mathbf{H}^{(t)}$ and $\mathbf{E}^{(t)}$, class label embeddings $\mathbf{R}^{(t)}$ are semantically dependent and trainable. This implies they can capture and leverage more information. To the best of our knowledge, it is the first attempt to explicitly design the model to accommodate incremental labels in the online cross-modal hashing domain.

## 3.3 Hash-Code Learning

After the first step of our method, we have acquired label embeddings. In certain Hadamard matrix-based approaches [17, 19], the Hadamard matrix is directly applied to learn hash functions. In contrast, our method employs the learned label embeddings to generate hash codes for the $t$-th round of training samples. In the context of online scenarios, when new data arrives, the online cross-modal hashing methods focus on generating new hash codes for these data while maintaining the binary codes of previously observed streaming data unchanged. However, there are two key challenges: 1) The distribution of newly arriving data can change, particularly with the emergence of new classes. Solely learning from new data may result in catastrophic forgetting. 2) The quality of incoming data is uncertain, and poor-quality data might negatively impact hash code learning. To address these challenges, we introduce globally invariant information, the learned label embeddings. These embeddings contain semantic information and class correlation and can serve as guidance for producing hash codes of new data. Unlike the feature space, the label space undergoes minimal changes with the influx of streaming data. We generate embeddings for fundamentally invariant labels and transfer information from the original data blocks to the new ones by measuring the similarity between the hash codes of samples and label embeddings. Consequently, we further define the following optimization problem,

$$\min_{\vec{\mathbf{B}}^{(t)}} \beta \parallel r\vec{\mathbf{L}}^{(t)} - \mathbf{R}^{(t)\top}\vec{\mathbf{B}}^{(t)} \parallel_F^2 + \mathscr{L}(\vec{\mathbf{B}}^{(t)}), \quad s.t. \ \vec{\mathbf{B}}^{(t)} \in \{-1, 1\}^{r \times n_t}. \tag{10}$$

Our approach serves as a plug-in to enhance the performance of the original method, where $\mathscr{L}(\vec{\mathbf{B}}^{(t)})$ represents the loss function used by the original method to learn $\vec{\mathbf{B}}^{(t)}$. Furthermore, once the hash codes are learned, our method does not delve into the learning of hash functions. Our approach directly employs the learning methods described in the original text to obtain hash functions and uses them to generate hash codes for out-of-sample data.

## 3.4 Optimization Algorithm

### 3.4.1 Optimization for the Label Embedding.
In this step of our method, we aim to learn label embeddings by optimizing Eq.(9). Concretely, we propose the following two-stage iterative optimization to solve $\vec{\mathbf{R}}^{(t)}$ and $\mathbf{W}^{(t)}$. In each stage, one variable is updated with others fixed.

**Update $\vec{\mathbf{R}}^{(t)}$.** By setting the derivative of Eq.(9) w.r.t. $\vec{\mathbf{R}}^{(t)}$ to zero, we can update it by,

$$\vec{\mathbf{R}}^{(t)} = (\tilde{\mathbf{H}}^{(t)}\tilde{\mathbf{H}}^{(t)\top} + \vec{\mathbf{H}}^{(t)}\vec{\mathbf{H}}^{(t)\top} + \mathbf{I} + \alpha\mathbf{W}^{(t)}\mathbf{W}^{(t)\top})^{-1}$$
$$\cdot (r\tilde{\mathbf{H}}^{(t)}\mathbf{A}_{no}^{(t)} + r\vec{\mathbf{H}}^{(t)}\mathbf{A}_{nn}^{(t)} + \alpha\vec{\mathbf{H}}^{(t)} + \alpha\mathbf{W}^{(t)}\vec{\mathbf{E}}^{(t)\top}), \tag{11}$$

where $\mathbf{I} \in \mathbb{R}^{r \times r}$ is the identity matrix.

**Update $\mathbf{W}^{(t)}$.** With $\vec{\mathbf{H}}^{(t)}$ fixed, we could directly take the derivative of Eq.(8) w.r.t. $\mathbf{W}^{(t)}$ to zero and the solution for $\mathbf{W}^{(t)}$ can be obtained as shown below,

$$\mathbf{W}^{(t)} = \left(\tilde{\mathbf{R}}^{(t)}\tilde{\mathbf{R}}^{(t)\top} + \vec{\mathbf{R}}^{(t)}\vec{\mathbf{R}}^{(t)\top}\right)^{-1} \left(\tilde{\mathbf{R}}^{(t)}\tilde{\mathbf{E}}^{(t)\top} + \vec{\mathbf{R}}^{(t)}\vec{\mathbf{E}}^{(t)\top}\right). \tag{12}$$

### 3.4.2 Optimization for Hash Code Learning.
In this step of our method, we only need to learn one variable $\vec{\mathbf{B}}^{(t)}$ at round t.

**Update $\vec{\mathbf{B}}^{(t)}$.** By matrix operations and omitting constant terms in Eq. (10), we can get the following formula to learn $\vec{\mathbf{B}}^{(t)}$,

$$\max_{\vec{\mathbf{B}}^{(t)}} tr\left(\beta(r\mathbf{R}^{(t)}\vec{\mathbf{L}}^{(t)})\vec{\mathbf{B}}^{(t)\top}\right) + \mathscr{L}(\vec{\mathbf{B}}^{(t)}), \tag{13}$$

For Eq.(13), it is easy to find its closed-form solution,

$$\vec{\mathbf{B}}^{(t)} = sign(\beta\lambda r\mathbf{R}^{(t)}\vec{\mathbf{L}}^{(t)} + \vec{\mathbf{B}}_o^{(t)}), \tag{14}$$

where $\vec{\mathbf{B}}_o^{(t)}$ represents the hash codes optimized using the loss function described in the original paper. To enhance the adaptability of POLISH and combine it with other methods, we introduced a parameter $\lambda$, which aims to balance the magnitude of the loss function. Specifically, $\lambda$ is set to $sum(abs(r\mathbf{R}^{(t)}\vec{\mathbf{L}}^{(t)}))/sum(abs(\vec{\mathbf{B}}_o^{(t)}))$. Then, $\beta$ is a balancing parameter that measures the importance of POLISH relative to the original methods.

### 3.4.3 Discussions.
We give the computational complexity analysis of our method. We can find that the size of $\mathbf{A}^{(t)}$, $\mathbf{R}^{(t)}$, and $\mathbf{H}^{(t)}$ is $C_t \times C_t$, $r \times C_t$, and $r \times C_t$, respectively. Thus, the complexity of Eq.(8) is irrelevant to the size of the new data chunk, i.e., $n_t$. Specifically, at round $t$, the computational complexity of updating $\mathbf{R}^{(t)}$ is $O((c_t r^2 + C_t r^2 + d_e r^2 + C_t c_t r + c_t^2 r + r^3 + r d_e c_t) iter)$. Updating $\mathbf{W}^{(t)}$ requires $O((c_t r^2 + C_t r^2 + C_t d_e + c_t d_e r + r^3) iter)$. The computational complexity of updating $\vec{\mathbf{B}}^{(t)}$ is $O(c_t r n_t)$. Here, $iter$ stands for the number of iterations. It can be observed that the computational complexity of our method is linearly related to the size of new data $n_t$, demonstrating the scalability of our method.

## 4 EXPERIMENT

To evaluate the performance of POLISH, we carried out comprehensive experiments on two real-world datasets with the aim of addressing the following questions: 1) How well does POLISH perform when confronted with the introduction of new classes? 2) Does POLISH effectively work, and if so, why? In this section, we first introduce the dataset and experimental settings. Subsequently, we present the experimental results and provide an analysis for each of the aforementioned research questions.

## 4.1 Datasets

We conducted extensive experiments on two widely-used benchmark datasets: MIRFlickr [10] and NUS-WIDE [2].

**MIRFlickr** comprises $25,000$ images sourced from Flickr. Each image is associated with $1,386$ user-provided tags and is categorized into 24 classes. Similar to [12], tags appearing fewer than 20 times are eliminated, leaving $20,015$ instances. **NUS-WIDE** is a substantial dataset containing $269,648$ images, sourced from Flickr by the Lab for Media Search at the National University of Singapore. For our experiments, we selected the 21 most frequent labels, resulting in $195,834$ images, each associated with a $1,000$-dimensional binary tagging vector. In both datasets, we employed the $4,096$-dimensional output of the pre-trained VGG-F modal [1], which is initially trained on the ImageNet dataset, to represent images.

**Table 1: Cross-modal retrieval for class incremental data in online scenarios. The MAP results of various methods on MIRFlickr and NUS-WIDE at the last chunk. The best MAP values of each case are shown in boldface.**

| Dataset | Method | Image-to-Text | | | | Text-to-Image | | | |
|---|---|---|---|---|---|---|---|---|---|
| | | 32 bits | 64 bits | 96 bits | 128 bits | 32 bits | 64 bits | 96 bits | 128 bits |
| MIRFlickr | DCH [38] | 0.7239 | 0.7511 | 0.7478 | 0.7700 | 0.7097 | 0.7507 | 0.7462 | 0.7703 |
| | LCMFH [32] | 0.7859 | 0.7990 | 0.7957 | 0.8055 | 0.7562 | 0.7655 | 0.7650 | 0.7749 |
| | SCRATCH [14] | 0.7884 | 0.8040 | 0.8038 | 0.8084 | 0.7141 | 0.7135 | 0.7165 | 0.7170 |
| | LFMH [45] | 0.7312 | 0.7502 | 0.7756 | 0.7979 | 0.7067 | 0.7325 | 0.7589 | 0.7761 |
| | TASPH [8] | 0.7993 | 0.8143 | 0.8272 | 0.8325 | 0.7433 | 0.7549 | 0.7623 | 0.7529 |
| | OCMFH [33] | 0.5191 | 0.5334 | 0.5192 | 0.5217 | 0.5182 | 0.5349 | 0.5185 | 0.5220 |
| | LEMON [35] | 0.7906 | 0.7922 | 0.7991 | 0.7954 | 0.6981 | 0.7130 | 0.7163 | 0.7200 |
| | LEMON+POLISH | **0.8492** | **0.8389** | **0.8622** | **0.8295** | **0.8247** | **0.8282** | **0.8440** | **0.8090** |
| | DOCH [44] | 0.8269 | 0.8535 | 0.8565 | 0.8540 | 0.7788 | 0.8020 | 0.8044 | 0.8220 |
| | DOCH+POLISH | **0.8343** | **0.8680** | **0.8768** | **0.8592** | **0.8012** | **0.8278** | **0.8400** | **0.8382** |
| | LSE-OH [13] | 0.7906 | 0.7922 | 0.7991 | 0.7954 | 0.6981 | 0.7130 | 0.7190 | 0.7391 |
| | LSE-OH+POLISH | **0.8379** | **0.8512** | **0.8639** | **0.8438** | **0.7398** | **0.7540** | **0.7701** | **0.7498** |
| NUS-WIDE | OCMFH [33] | 0.5191 | 0.5334 | 0.5192 | 0.4303 | 0.5182 | 0.5349 | 0.5185 | 0.4295 |
| | LEMON [35] | 0.8544 | 0.7974 | 0.8000 | 0.7914 | 0.8270 | 0.7883 | 0.7829 | 0.7753 |
| | LEMON+POLISH | **0.8662** | **0.8691** | **0.8883** | **0.8846** | **0.8415** | **0.8642** | **0.8440** | **0.8622** |
| | DOCH [44] | 0.8847 | 0.8919 | 0.8955 | **0.8969** | 0.8547 | 0.8615 | 0.8649 | 0.8660 |
| | DOCH+POLISH | **0.8985** | **0.8969** | **0.8993** | 0.8936 | **0.8599** | **0.8807** | **0.8841** | **0.8758** |
| | LSE-OH [13] | 0.8781 | 0.8502 | 0.8505 | 0.8315 | 0.7838 | 0.7628 | 0.7644 | 0.7497 |
| | LSE-OH+POLISH | **0.8795** | **0.8768** | **0.8870** | **0.8880** | **0.7885** | **0.7990** | **0.8128** | **0.8031** |

To construct class-incremental scenarios, we shuffled the order of samples in both datasets. More specifically, the datasets, MIRFlickr and NUS-WIDE, were divided into 10 and 20 chunks, respectively. In each chunk, a certain proportion of samples introduced new classes (labels not seen in previous rounds). Additionally, 10% of the image-text pairs were randomly selected as the query set, while the rest formed the training set in each chunk.

## 4.2 Experimental Settings

### 4.2.1 Evaluation Metrics and Parameter Setting.
The evaluation of POLISH encompasses two cross-modal retrieval tasks: 1) In "Image-to-Text" task, an image serves as a query to retrieve relevant text data. 2) "Text-to-Image" task uses a text as a query to retrieve relevant images. Similar to previous methods, we employ a widely used evaluation criterion for assessing performance, namely the Mean Average Precision (MAP). A higher MAP value signifies superior performance.

The parameters of POLISH are selected experimentally. We set $\alpha = 10$, $\beta = 10$ on both MIRFlickr and NUS-WIDE, more details are shown in Sec. 4.4.3. And the iteration number of learning label embeddings is set to 5.

### 4.2.2 Baselines.
To validate the efficacy of our proposed method, we conducted a comprehensive comparative analysis against several state-of-the-art cross-modal hashing baselines, which encompass five batch-based approaches, namely DCH [38], LCMFH [32], SCRATCH [14], LFMH [45], and TASPH [8], in addition to four online methods, OCMFH [33], LEMON [35], DOCH [44], and LSE-OH [13]. Notably, OCMFH is categorized as an unsupervised online cross-modal hashing method, whereas others are supervised ones.

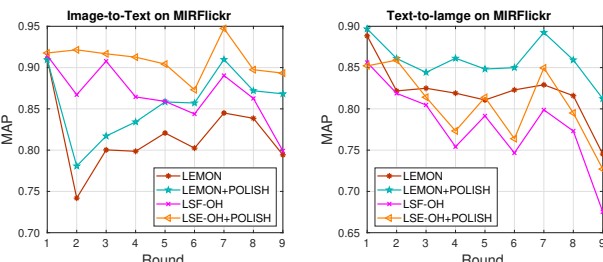

**Figure 2: Cross-modal retrieval in online scenarios. The MAP-round curves of several methods on MIRFlickr with 64 bits.**

For all batch-based baselines, during each round, all available data is aggregated for the training of their hash functions and hash codes. The source code for these baselines is publicly accessible.

It is worth highlighting our choice of these three works as supervised online cross-modal hashing baselines. Here's why: 1) LEMON is a classical method that optimizes hash codes using an auxiliary variable strategy. It predominantly focuses on the relevance among newly arrived data or between new data and the existing data for hash code learning while overlooking valuable global semantic information. 2) DOCH is a typical method that employs discrete techniques to optimize hash codes. It involves the random selection of anchors from the old data to preserve previous knowledge. This strategy has the same propose as ours, which can effectively address the problem of catastrophic forgetting. 3) LSE-OH is the most similar method to ours as it also learns label representations.

**Table 2: The results of ablation experiments on MIRFlickr. The best MAP values of each chunk are shown in boldface**

| Method | Image-to-Text | | | | Text-to-Image | | | |
|---|---|---|---|---|---|---|---|---|
| | 32 bits | 64 bits | 96 bits | 128 bits | 32 bits | 64 bits | 96 bits | 128 bits |
| LSE-OH | 0.7906 | 0.7922 | 0.7991 | 0.7954 | 0.6981 | 0.7130 | 0.7163 | 0.7200 |
| LSE-OH+POLISH-1 | 0.7830 | 0.8145 | 0.8077 | 0.8160 | 0.6908 | 0.7313 | 0.7249 | 0.7364 |
| LSE-OH+POLISH-2 | 0.7964 | 0.8138 | 0.8145 | 0.7974 | 0.7245 | **0.7566** | 0.7603 | 0.7505 |
| LSE-OH+POLISH-3 | 0.8098 | 0.8418 | 0.8496 | 0.8354 | 0.7189 | 0.7454 | 0.7522 | 0.7496 |
| LSE-OH+POLISH-4 | 0.7984 | 0.8490 | 0.8591 | **0.8574** | 0.7231 | 0.7475 | 0.7616 | **0.7544** |
| LSE-OH+POLISH | **0.8379** | **0.8512** | **0.8639** | 0.8438 | **0.7398** | 0.7540 | **0.7701** | 0.7498 |

However, there are notable distinctions. LSE-OH regenerates label representations in each training round, while in POLISH, label representations are learned once and remain constant to provide global guidance. Additionally, the label representations in LSE-OH are binary codes, not real values, resulting in information loss.

## 4.3 Comparison When New Classes Come (Q1)

To the best of our knowledge, this is the first work addressing the issue of incremental label space in the domain of online cross-modal hashing tasks. In order to evaluate our approach, we reshuffled the order of samples in the datasets to simulate class-incremental scenarios. In each round, the new data consistently introduces class labels that are not previously seen. Supervised methods often exhibit poor performance and even fail to work in scenarios with label increments. This is primarily due to potential issues concerning mismatches in label matrix dimensions between different rounds. Consequently, for all methods except POLISH, we zero-padded the data labels to ensure label dimension consistency.

To evaluate POLISH, we presented the MAP results for the final round on MIRFlickr and NUS-WIDE in Table 1. Moreover, for a comprehensive portrayal of online retrieval performance, Figure 2 provides an illustrative presentation of MAP results for several methods across each round, taking into account 64-bit representations. Analysis of these outcomes reveals that:

- Our proposed method, POLISH, consistently demonstrates a substantial enhancement in performance over the original baselines under the class-incremental scenarios in both benchmark datasets. For instance, when the hash code length is fixed at 64 bits, POLISH outperforms the LSE-OH method by a remarkable 5.9% and 4.1% in the "Image-to-Text" and "Text-to-Image" tasks on MIRFlickr.
- We also compared our method with some batch-based supervised cross-modal hashing methods on the MIRFlickr. These batch-based methods aggregate all available data in each round to train hash functions and codes. It is evident that online methods consistently outperform batch-based methods. One possible reason is that in the streaming data scenario, online methods often consider the similarity between new and old data, which helps learn more accurate hash codes.
- Thanks to its well-designed module, POLISH demonstrates remarkable performance when dealing with the arrival of new classes. In comparison to LEMON, which ignores global information, POLISH exhibits significant improvements. This indicates that in class-incremental scenarios, global information plays a guiding role and can alleviate catastrophic forgetting.

- Combining the discrete optimization method DOCH with POLISH, DOCH+POLISH consistently outperforms DOCH in all cases. This suggests that merely selecting anchors from old data to retain prior knowledge is insufficient, and employing invariant label representations as global information yields superior results.
- In contrast to LSE-OH, LSE-OH+POLISH achieves superior performance. This implies that using binary label representations learned from label co-occurrence may lead to the loss of valuable information. POLISH leverages the correlation and semantic information of labels obtained from language models to learn real-value representations and gat better results.
- The MAP-round curves on two datasets demonstrate a similar trend, which reveals the effectiveness of our method.

From the above results, we can conclude that POLISH performs exceptionally well. It benefits not only from utilizing language models to learn label relevance and semantic information but also from using obtained label representations as global information to guide the learning of hash codes.

## 4.4 Further Analysis (Q2)

*4.4.1 **Ablation Experiments**.* To verify the effectiveness of our method, we designed several variants. 1) POLISH-1: it omits the first three terms in Eq. (9), meaning that we only used the semantic information obtained by language models when learning label representations, neglecting the correlation among labels. 2) POLISH-2: it sets $\alpha = 0$, which only considers the correlations among labels when learning label representations. 3) POLISH-3: it learns binary embedding $\mathbf{R}^{(t)}$ of labels rather than real-valued $\mathbf{R}^{(t)}$. 4) POLISH-4: It does not use Hadamard matrices to assist in learning $\mathbf{R}^{(t)}$. The MAP results of these variants combined with the LSE-OH method are presented in Table 2. From this table, we can find:

- LSE-OH+POLISH outperforms LSE-OH+POLISH-1, demonstrating that by further preserving correlation among labels, better label representations can be obtained.
- LSE-OH+POLISH always offers better performance compared with LSE-OH+POLISH-2, revealing the superiority of embedding the semantic information into label representations.
- LSE-OH+POLISH-3 is worse than LSE-OH+POLISH in most cases. One possible reason is that the real-valued embedding could preserve more accurate information.
- LSE-OH+POLISH performs better than LSE-OH+POLISH-4 in most cases, revealing the effectiveness of incorporating Hadamard matrices into label embedding learning.

Table 3: The performance of the LEMON+POLISH method using various language models on MIRFlickr. The best MAP values are highlighted in bold.

| Method | Image-to-Text | | | | Text-to-Image | | | |
|---|---|---|---|---|---|---|---|---|
| | 32 bits | 64 bits | 96 bits | 128 bits | 32 bits | 64 bits | 96 bits | 128 bits |
| Word2Vec [25] | 0.8075 | **0.8478** | 0.8516 | 0.8262 | 0.7813 | **0.8256** | 0.8303 | 0.7971 |
| BERT [3] | 0.8105 | 0.8250 | **0.8647** | 0.8373 | 0.7837 | 0.7932 | 0.8192 | 0.7846 |
| CLIP [27] | **0.8220** | 0.8404 | 0.8262 | 0.8089 | **0.8056** | 0.8066 | 0.7963 | 0.7761 |
| RoBERTa [21] | 0.8034 | 0.8059 | 0.8456 | 0.8427 | 0.7696 | 0.7856 | 0.8257 | 0.7885 |
| BLOOM [28] | 0.7741 | 0.8306 | 0.8458 | **0.8463** | 0.7697 | 0.8079 | **0.8383** | **0.8071** |

- POLISH and all its variants outperform LSE-OH, demonstrating the effectiveness of our designed plug-in. By harnessing multi-angle information provided by language models, we have learned valuable label embeddings that represent global information. Simultaneously, these embeddings guide the learning process, enabling POLISH to retain more information and generate more accurate hash codes.

### 4.4.2 Language Models Analysis.
Language models have made significant strides in recent years, especially in the past few months, and have garnered considerable attention. Models like BERT, GPT, and others exhibit powerful natural language processing capabilities, finding applications across various domains. In this paper, we leveraged language models to explore the latent knowledge encapsulated within labels. Using the robust representational capabilities of language models, we generated label correlation and semantic information and utilize them to generate learnable label embeddings. To further explore the performance of our approach, we evaluated it using different language models. Specifically, we utilized Word2Vec [25], BERT [3], CLIP [27], RoBERTa [21], and BLOOM [28]. The MAP values they achieve on the MIRFlickr dataset are presented in Table 3. From this table, we could have the following observations:

- Our method consistently performs well, regardless of the language model used. This demonstrates the versatility and adaptability of our approach.
- When the hash code length is relatively high, some excellent large language models like BLOOM perform well. However, due to their high-dimensional representations (BLOOM has a representation dimension of 2560), when using them to generate low-dimensional hash codes of samples, a significant amount of information is lost, resulting in performance that falls short of Word2Vec and RoBERTa.
- In low-dimensional scenarios, CLIP performs well. One possible reason is that CLIP is the only model that aligns image and text modalities during training. This natural alignment is well-suited for cross-modal retrieval tasks.

The focus of this paper is not to determine which language model might perform better. We are primarily proposing the potential of integrating online cross-modal retrieval tasks with language models. Therefore, in this paper, we used Word2Vec as the language model for extracting information in other experiments.

### 4.4.3 Sensitivity to Parameters.
We conducted experiments on MIRFlickr and NUS-WIDE to analyze the influence of parameters on the performance in the case of 32-bit code length. The results

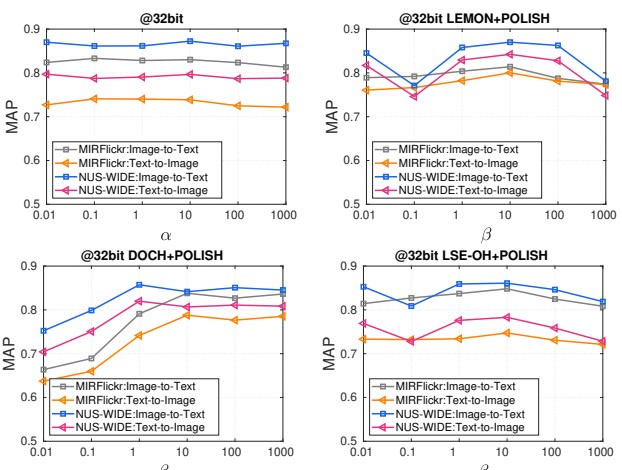

Figure 3: Parameter sensitivity analysis of $\alpha$ and $\beta$.

on MIRFlickr and NUS-WIDE are presented in Figure 3. We exclusively evaluated the performance impact of different $\alpha$ values on the LSE-OH method, with similar results expected for other methods. Notably, POLISH demonstrates robustness to the $\alpha$, performing well across a broad range from $10^{-2}$ to $10^3$. Conversely, $\beta$ governs the balance between POLISH and the original baselines, the sensitivity of the $\beta$. The value of $\beta$ can significantly affect the results, contingent on the specific method. Consequently, we presented the performance of the $\beta$ across all methods. While $\beta$ proves to be sensitive, it consistently yields favorable results when set to 10 across all datasets. This observation underscores the critical role of our plugin in hash code learning. In conclusion, through the experiment, we set $\alpha = 10$, $\beta = 10$ on both MIRFlickr and NUS-WIDE.

## 5 CONCLUSION

In this paper, we introduce a novel supervised online cross-modal hashing method, referred to as adaPtive Online cLass- Incremental haSHing (POLISH). To the best of our knowledge, this is the first attempt to explicitly design a model to accommodate incremental label spaces in the online cross-modal hashing domain. POLISH utilizes language models to learn label embeddings and translates class correlation and class semantic information into label embeddings. Besides, POLISH uses class embeddings as global information to guide hash code learning, thereby enhancing hash code accuracy. Extensive experiments were conducted on real-world benchmark datasets, and the results demonstrate the superiority of POLISH.

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
