# OpenReview forum: "POLISH: Adaptive Online Cross-Modal Hashing for Class Incremental Data"
_ACM.org/TheWebConf/2024/Conference — TheWebConf24 Oral_

### Official Review · Reviewer_i6cM · 2023-11-19

**Novelty:** 4
**Technical Quality:** 4

**Review:**

This paper introduces an approach named POLISH for adaptive online cross-modal hashing. The authors highlight the importance of preserving label correlation relationships and propose a loss function to learn the embedding of class labels. Experimental results show that this method has better performance than the baselines.

**Paper Strengths**

The topic itself is very interesting. This paper is generally well-written and organized, making it easy to follow. The authors provide clear explanations of the concepts and techniques used in the proposed method.

**Paper Weaknesses**

1. This paper could benefit from further clarification of the notation used. It is inappropriate to use ~ and -> superscripts to distinguish matrix notations. Moreover, there are too many inline formulas used in the paper.
2. More detailed explanations of the experimental setup and experiments would support the proposed method. See details in Question.

**Questions:**

1. Would the proposed method still work if a round in training occurs when there are a large number of new classes, even much more than the exsiting ones? This is very likely to happen, especially at the beginning of the training, which is not considered in the paper and there are no corresponding experiments. In addition, does the size of data volume in a new round during training have an impact on the results? This is also an issue that needs to be considered.
2. Why are the baselines of the NUS-WIDE dataset different from MIRFlickr in Table 1?
3. There were only two datasets used in the experiment. Does it still work on other datasets? For example, the MSCOCO dataset was used in OCMFH.

**Reviewer Confidence:**

2: The reviewer is willing to defend the evaluation, but it is likely that the reviewer did not understand parts of the paper

**Scope:**

3: The work is somewhat relevant to the Web and to the track, and is of narrow interest to a sub-community

---

### Official Review · Reviewer_Kp2U · 2023-11-21

**Novelty:** 3
**Technical Quality:** 3

**Review:**

The paper introduces a supervised online cross-modal hashing method named adaPtive Online cLass Incremental haSHing (POLISH) method. It does not rely on the assumption that the label space remains constant.

**Questions:**

Advantage

1. The topic of the paper is interesting. Cross-modal retrieval and online settings are useful and important in real-world applications.
2. The paper applies language models to guide model learning. It is a reasonable method.
3. In experiments, the proposed method outperforms baselines.

Disadvantage

1. The motivation should be further explained. The problem is not formulated clearly. Although the paper has a problem description in Section 3.1.2, the paper does not explain the input and output of the model. And based on the paper, it seems to learn the hash codes for each data directly instead of leveraging a deep neural network. However, deep multi-modality models such as CLIP can produce high-quality representation. If we add a hash layer at the top of CLIP to achieve hash codes, it seems the online scenario is not a big issue.
2. The usage of pre-trained language model is confusing. Language models should have context to generate meaningful embeddings. However, the paper just uses label words. Additionally, computing the similarity using the embedding from language models is not effective [1]. And this is the reason that using a more powerful language model does not result in better performance. Furthermore, it is not reasonable to use BLOOM, which is a generative model and not good at generating good representation.
3. The writing should be improved. First, there are multiple typos, such as line 354 \tilde{E}. Second, some technical details should be explained. For example, the meaning of line 496 “loss function used by the original method” should be explained. Third, There are many colloquial expressions。


[1] Reimers N, Gurevych I. Sentence-bert: Sentence embeddings using siamese bert-networks[J]. arXiv preprint arXiv:1908.10084, 2019.

**Reviewer Confidence:**

4: The reviewer is certain that the evaluation is correct and very familiar with the relevant literature

**Scope:**

4: The work is relevant to the Web and to the track, and is of broad interest to the community

---

### Official Review · Reviewer_7bam · 2023-11-21

**Novelty:** 4
**Technical Quality:** 6

**Review:**

The study proposes a method to create hash-code representations of high-dimensional streaming data(images and text) to be used in ANN search methods. It is targeted for online(streaming) cross-modal retrievals.

The main contributions are:
- able to work with incremental label spaces.
- designed to be used as a plug-in to any cross-modal hashing methods.
- makes use of LLMs to create embeddings for new labels, and then transforms them into hash code.
- efficient in terms of time-complexity.

Some weak areas:
- some choices are not very clear. In the related work they introduce the Hadamard matrices, but do not explore alternative, if they exist.
- other clarification points in questions.

**Questions:**

1. Regarding the ablation studies, POLISH-2 is better than POLISH in one case, why is that? Why does not considering the correlation improve the performance? Could it have a relation with the number of labels or their properties?

2. How would this method work without the Hadamard matrices? As far as I understand, it is due to these matrices that the R(t) representation has the bit balance and maximal information properties. Would the method work correctly without such properties?

3. It is mentioned that the method grows in time linearly with regards to the size of the newly arrived data. (q5) Will this drastically affect the training time of methods that grow linearly themselves(e.g. LEMON)?

4. The smallest representations you have experimented with have 32 bits. Was there a reason you did not look into smaller ones(8 or 16)?

5. Small typo in Equation (1), it minimizes for R(1)

6. The POLISH-3 always performs worse, typo I think

**Ethics Review Description:**

no issue

**Reviewer Confidence:**

2: The reviewer is willing to defend the evaluation, but it is likely that the reviewer did not understand parts of the paper

**Scope:**

4: The work is relevant to the Web and to the track, and is of broad interest to the community

---

### Official Review · Reviewer_feBM · 2023-11-23

**Novelty:** 6
**Technical Quality:** 6

**Review:**

**Quality and Clarity:**
- The paper are clearly written, succinctly explaining the problem of adapting hashing-based retrieval systems to streaming web data. It then articulates the challenges in existing methods and proposes a solution with clarity.

**Originality:**
- The originality lies in the creation of the POLISH method, which adapts online cross-modal hashing to class incremental data. Incorporating category correlation and semantic information for hash code generation is an innovative aspect of this work.

**Significance:**
- The significance of this work is high, particularly in the context of real-time web data processing and retrieval.
- Offers potential improvement over batch-based cross-modal hashing methods, which is a substantial advancement in the field.

**Pros:**
- Addresses a crucial and timely issue in data retrieval with an innovative solution.
- POLISH is designed to update hash functions effectively, which is a challenge in current systems.
- Introduction of an efficient optimization algorithm for discrete learning of hash codes and label embeddings.
- Thorough experiments and comparison analysis demonstrate the effective of POLISH method in streaming data environments.

**Cons:**
- While the approach is novel, the efficiency compared to state-of-the-art methods is not discussed.

**Questions:**

What's the efficiency of POLISH compared with other methods? Is it scalable?

**Reviewer Confidence:**

3: The reviewer is confident but not certain that the evaluation is correct

**Scope:**

4: The work is relevant to the Web and to the track, and is of broad interest to the community

---

### Official Review · Reviewer_jB6V · 2023-11-23

**Novelty:** 4
**Technical Quality:** 5

**Review:**

POLISH presents a novel and technically sophisticated solution to the problem of incremental label spaces in online cross-modal hashing. The incorporation of language models and the strategic use of embeddings for global guidance are commendable technical strengths. To enhance the paper’s technical depth, further exploration of language models and a more extensive comparative analysis could be beneficial.

**Questions:**

Strength:
	Online cross-modal hashing involves learning representations (hash codes) for data instances, such as images and text, in an incremental and online manner. This means the system adapts to new data as it arrives, updating its knowledge without retraining on the entire dataset.
	POLISH presents a distinctive viewpoint by explicitly taking into account situations where the label space undergoes incremental changes. This deviates from conventional approaches, which typically assume a static set of labels. The system dynamically adjusts to the incorporation of new classes in each learning iteration, showcasing a forward-thinking approach for real-world applications.
	An essential technical advancement in POLISH involves incorporating language models such as Word2Vec, BERT, and CLIP to tap into latent knowledge embedded in labels. Through utilization of these models, POLISH produces valuable label embeddings that encompass both semantic details and correlations among labels. This methodology surpasses conventional hashing techniques, enhancing the creation of more comprehensive representations.

Weakness:
	While the paper compares POLISH against several state-of-the-art methods, a more in-depth comparative analysis, especially against methods addressing incremental label spaces, could further highlight the unique advantages of POLISH. This would strength the argument for its novelty in addressing this specific challenge.
	While POLISH demonstrates robust performance across different language models (Word2Vec, BERT, CLIP, etc.), the paper does not extensively explore the impact of different models on the system’s performance. Further investigation into the choice of language model and its implications could enhance the understanding of POLISH’S versatility.
	The sensitivity of the system to the β parameter is acknowledged, and the paper reports consistent favorable results when β is set to 10. However, a deeper exploration of the sensitivity across a broader range of values and its implications on performance could provide more insights into the system’s behavior under different settings.

**Ethics Review Description:**

-

**Reviewer Confidence:**

4: The reviewer is certain that the evaluation is correct and very familiar with the relevant literature

**Scope:**

3: The work is somewhat relevant to the Web and to the track, and is of narrow interest to a sub-community

---

### Decision · Program_Chairs · 2024-01-22

**Decision:**

Accept (Oral)

**Comment:**

Overall, the reviewers recognize substantial novelty and potential for impact. The authors have engaged constructively in the reviewing process. In some cases the reviewers have acknowledged responses, and in any case, I believe most questions raised by the reviewers have been adequately addresses.

 One reviewer in particular gives relatively low scores for novelty and technical quality. Their concerns are mostly presentational, and not anything that would block acceptance. The authors have responded to these concerns. While the reviewer did not comment further, in my opinion the authors have adequately addressed the concerns.

 Below, I recommend oral presentation because I believe the paper would be interesting to a relatively broad range of attendees. However I am not well calibrated on the breakpoint between oral and poster presentation.